# Potential Future Alternative Resources for Rare Earth Elements: Opportunities and Challenges

**Vysetti Balaram**

CSIR—National Geophysical Research Institute, Hyderabad 500 007, India; balaram1951@yahoo.com

**Abstract:** Currently, there is an increasing industrial demand for rare earth elements (REE) as these elements are now integral to the manufacture of many carbon-neutral technologies. The depleting REE ores and increasing mining costs are prompting us to consider alternative sources for these valuable metals, particularly from waste streams. Although REE concentrations in most of the alternative resources are lower than current REE ores, some sources including marine sediments, coal ash, and industrial wastes, such as red mud, are emerging as promising with significant concentrations of REE. This review focuses on the alternative resources for REE, such as ocean bottom sediments, continental shelf sediments, river sediments, stream sediments, lake sediments, phosphorite deposits, industrial waste products, such as red mud and phosphogypsum, coal, coal fly ash and related materials, waste rock sources from old and closed mines, acid mine drainage, and recycling of e-waste. Possible future Moon exploration and mining for REE and other valuable minerals are also discussed. It is evident that REE extractions from both primary and secondary ores alone are not adequate to meet the current demand, and sustainable REE recovery from the alternative resources described here is also necessary to meet the growing REE demand. An attempt is made to identify the potential of these alternative resources and sustainability challenges, benefits, and possible environmental hazards to meet the growing challenges of reaching the future REE requirements.

**Keywords:** rare earth elements; deposits; alternative sources; marine sediments; river sediments; phosphorites; red mud; fly ash; acid mine drainage; e-waste; extra-terrestrial





## 1. Introduction

Seventeen elements in the periodic table including fifteen lanthanides (La, Ce, Pr, Nd, Pm, Sm, Eu, Gd, Tb, Dy, Ho, Er, Tm, Yb, and Lu) and Sc, as well as Y are often collectively referred to as rare earth elements (REE). Promethium is well-known as the only element in the lanthanide series of the periodic table with no stable isotopes and occurs in the Earth's crust in only tiny amounts in some natural materials, such as some uranium ores. Scandium is geochemically only partly similar to REE. Based on atomic numbers, they are divided into two groups. The lower atomic weight elements from La to Sm, the most abundant ones, with atomic numbers 57–62 are referred to as light REE (LREE), while from Eu to Lu, the least common and the most valuable with atomic numbers 63–71, are known as heavy REE (HREE). In nature, REE do not occur as single native metals, such as gold or silver since they are easily oxidized due to their similar physical and chemical properties. Moreover, they frequently occur together in several geologic formations in many ores or minerals as minor or major constituents.

Due to their unique physical, chemical, mechanical, electronic, magnetic, luminescent, phosphorescent, and catalytic properties, these elements have become exceptionally important during the last two decades, and there has been an explosion in the industrial applications of these elements in different high technology devices, such as smartphones, computers, televisions, light emitting diodes, catalysts for fuel cells, corrosion inhibitors, hard drives, magnets for wind turbines, and other power generating systems. Moreover,

they are a crucial element of national security as they are extensively used in several military defence systems [1–3]. Their important medical applications include metallic implants, lasers, and magnetic resonance imaging (MRI) measurements [4]. REE are vital, even to the space industry, as they are used in everything from launch vehicles to national defence and commercial communication and observation satellites. Furthermore, REE are used in creating catalysts in several industrial processes, as well as in the fabrication of autocatalytic catalytic converters in transport vehicles [5,6].

Although REE deposits are found practically all over the world, currently, there is a global shortage due to the decreasing number of economically profitable deposits. In addition, our reliance on REE in our high-tech gadget-hungry world is growing with time, but their supply is far from secure. Unfortunately, the conventional ore resources are becoming depleted, as the demand for REE increases, and the industry is paying attention to unconventional resources, such as coal, recycling, and marine sediments as alternative sources for these elements. Moreover, it is very important to maintain a proper supply chain to meet the demand for the development of highly advanced technologies. Even the substitution for the REE is difficult for most applications, although powerful permanent magnets made out of iron and nitrogen (iron nitride, $Fe_{16}N_2$) are helping the automotive and wind energy industries in a small way [7]. Pavel et al. [8] studied the possibilities of the substitution of REE in wind turbines in order to reduce the dependency on REE. However, later, it was found that these claims were impossible without neodymium. As a result, recycling is presently increasing in several countries. Many approaches including physical, chemical, and biological procedures, such as pyrometallurgy, solvent extraction, and membrane separation are available for the efficient extraction of REE from these resources [9]. One very important aspect is that currently extracted ore grades, with an average of 5% REE and running as high as 15%, have significantly higher concentrations than most of the discussed alternative sources [10]. Since only a few countries are producing these elements on a commercial scale, meeting the growing demand for these elements is becoming exceedingly difficult in several other countries. As the demand for REE is growing and conventional ores are becoming depleted, attention is turning to unconventional and alternative REE resources. Therefore, it is very important to understand the global REE resources, including alternative resources and their production scenario, and to take measures against future prospects. In this paper, I first present a comprehensive summary of the primary, secondary, and alternative REE resources and their potential to meet the demand. Then, I provide technical information for alternative REE resources other than the primary and secondary resources for policymakers and other key stakeholders. Before introducing the alternative REE resources, a short review of the major REE primary and secondary deposits is presented.

## 2. REE Deposits

Although REE comprise significant amounts of a wide range of minerals, including oxides, silicates, carbonates, phosphates, and halides, almost all production comes from less than ten minerals, such as apatite, monazite, xenotime, allanite, and bastnäsite [11]. Currently, these deposits are mainly located in China, Brazil, Vietnam, Canada, Russia, Namibia, South Africa, and India. Although China has been dominating in this area, several other countries, such as the USA, Australia, Turkey, and Sweden have successfully discovered new REE deposits recently. In fact, a number of new REE metallogenetic belts can be identified on the basis of age, tectonic setting, lithological association, and known REE enrichments [12]. REE deposits are divided into primary (formed by magmatic, hydrothermal, and/or metamorphic processes) and secondary deposits (formed by weathering and sedimentary transport) depending on their form of occurrence, genetic associations, and mineralogy [13]. Figure 1 presents an illustrative view of different types of REE resources, including the potential alternative future resources.

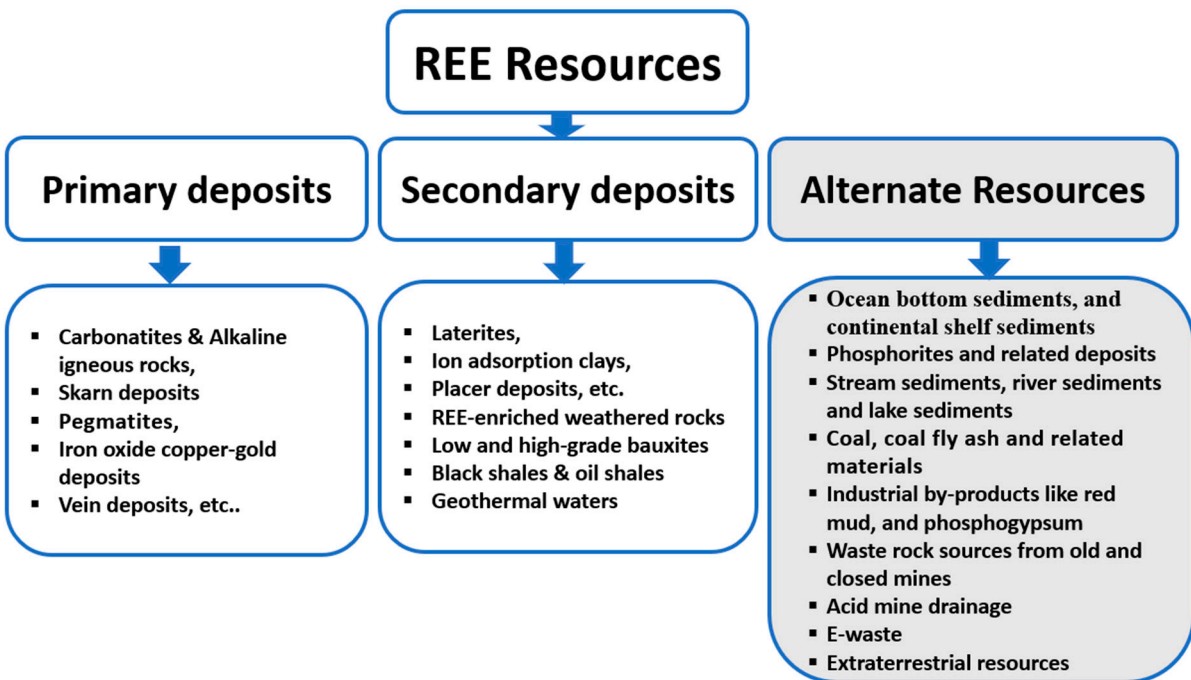

**Figure 1.** Illustrative view of different types of REE resources, including the potential alternative future resources shaded in light color [14].

### 2.1. Primary Deposits

Primary deposits of REE are commonly associated with carbonatites and alkaline igneous rocks, pegmatites, iron oxide copper-gold deposits, and vein and skarn deposits. Of these carbonatites, peralkaline silicate rocks with appreciable concentrations of REE are the most important REE resources [14,15]. These REE deposits in igneous rocks have played an important role in meeting the industrial demand for decades. Moreover, these igneous mineral deposits can be divided into five distinct categories depending on the provenance, the evolution of the magma, and the rock types hosting mineralization: (i) Carbonatites, (ii) peralkaline silica undersaturated rocks, (iii) peralkaline granites and pegmatites, (iv) pegmatites associated with sub- to meta-luminous granites, and (v) Fe oxide–phosphate deposits [16]. More insights on these primary REE deposits were presented by Weng et al. [17] and Dushyantha et al. [13]. Skarn deposits of REE are important in sustainable economic development across the world [18]. In Fe oxide–phosphate deposits, these elements are concentrated in apatite minerals [19].

### 2.2. Secondary Deposits

Some of the important secondary deposits of REE are laterites, ion adsorption clays, low- and high-grade bauxites, and placer deposits. Ion-adsorption deposits are formed due to the erosion and weathering of primary deposits within weathering crusts, which supply more than 95% of the global HREE demand. Intensive lateritic weathering of bedrocks under tropical or sub-tropical climatic conditions can form a variety of secondary deposits, which may range in composition from aluminous bauxites to hematite, goethite, and titanium sands, as well as REE [20]. Recently, Jo and Shin [21] reported REE-enriched weathered anorthosite rocks with ΣREE range of 242–857 µg/g. Ionic clays, formed by the natural weathering process of REE minerals and the adsorption of the resulting liberated REE ions on the clay surface, are an important resource for critical REE. These ion-adsorption clay deposits are most suitable for in situ leaching mining [22]. Ion-adsorption REE deposits are mostly developed due to weathering of REE-rich granites, and REE ions are mainly adsorbed on clay minerals [23]. Currently, regolith-hosted REE deposits from areas with abundant REE deposits have become one of the major sources of global REE. Regolith-

hosted REE deposits with abundant deposits around Meizhou City, Guangdong Province, China with the highest ΣREE range of 1162 µg/g were reported recently by Lin et al. [24]. The REE concentrations in the regolith-hosted REE deposits in the Chilean coastal range of the central Andes were found to be up to 2000 µg/g [25]. Beach sands are the products of a combination of weathering, fragmentation, and degradation, and are well-known for their economic concentration of heavy minerals, such as monazite, ilmenite, zircon, rutile, allanite, sillimanite, and garnet. REE are abundant in some of these minerals, particularly in monazite and allanite [26]. Black shales can also be considered as a secondary resource for these valuable metals. Recently, El-Anwar et al. [27] reported REE enrichment (average 255.3–325.3 µg/g) in black shales of the Safaga-Qussier sector, Egypt. Another study is reported from the rock-soil-moss system in the black shale area in China, with an average concentration of ∑REE around 245 µg/g [28]. Akhtar et al. [29] reported ∑REE concentrations up to 372 µg/g in Paleoproterozoic black shales from Singhbhum mobile belt, Eastern India. These recent studies, along with the industry standard of ~300 µg/g cut-off grade for mining, prompted more studies on black shale formations worldwide before considering the black shale deposits as an alternative source of REE. In fact, Ketris and Yudovich [30] reported an average worldwide REE concentration of 134.19 in black shales. Li et al. [31] reported moderately high concentrations of REE ranging from 105 to 195 µg/g at an average concentration of 151 µg/g in oil shale samples in Tongchuan City, Southern Ordos Basin, China, prompting the necessity for further investigations to understand the REE potential of these rocks. The value indicated above is significantly below the cut-off grade of 1000 µg/g (ash basis) for coal-hosted REE deposits [32]. In fact, the total REE contents, individual REE compositions, and sizes of different deposits are important for arriving at the cut-off value for the economic recovery of REE.

Geothermal fluids are potentially significant sources of valuable minerals and metals. Smith et al. [33] provided a rough estimate of total REE concentrations of 0.17 µg/mL in geothermal waters. Kurzawa et al. [34], in an attempt to understand the REE concentrations in the mineral and thermal waters in Polish Lowlands, found that a high temperature (T > 60 °C) favored the release of the more easily soluble REE from rocks into the water. Wei et al. [35] recently reported the total REE contents of the geothermal water in the range of 0.059–0.547 ng/mL, in the Ganzi–Litang fault, western Sichuan, China. The REE contents in geothermal waters are related to the reservoir lithology and are significantly influenced by pH and $HCO_3^-$, $Na^+$, and Mn minerals. Although the REE content in geothermal waters is not considerable, if the abundant geothermal waters in places, such as Iceland are taken into consideration, this source could be one of the most promising ones for REE with an efficient extraction technology. Recently, significant efforts have been made to develop technologies for the extraction of REE from geothermal brines since the geothermal brine can be a cost-effective mineral resource. REE extraction from geothermal brine only seems feasible at present if extracted with other co-products, such as silica and lithium [33].

### 2.3. Different Types of Potential Alternative Deposits

To date, the identified primary and secondary REE resources are not able to meet the global demand and requirements. Discovering and establishing alternative resources for REE have been a topic of high interest for the past decade. Therefore, a large number of studies are carried out for the identification of potential alternative deposits to meet the growing demand worldwide. Powerful analytical techniques, such as inductively coupled plasma mass spectrometry (ICP-MS), inductively coupled plasma optical emission spectrometry (ICP-AES), scanning electron microscopy with energy dispersive X-ray spectrometry (SEM-EDX), X-ray fluorescence spectrometry (XRF), X-ray diffractometry (XRD), and integrated mineral analysis (IMA), are being utilized to understand the future potential of various alternative REE resources [36–38]. In the following sections, I will present some of the most important alternative sources for REE.

### 2.3.1. Ocean Bottom Sediments and Continental Shelf Sediments

Rifting of the continents has formed widespread sedimentary basins, as rifting processes produce greater subsidence and lower basal heat flow, enlarging the depth extent of hydrothermal circulation and favoring the formation of giant deposits of elements, such as copper, lead, zinc, and nickel [39]. Recently, several studies revealed that marine sediments, such as ferromanganese crusts (cobalt crusts), manganese nodules, and marine mud from different parts of deep oceans are found to contain significant concentrations of REE and Y [40–43]. In particular, marine phosphorites are known to concentrate REE and Y during the early diagenetic formation, and the $\sum$REE+Y concentrations reach over 700 µg/g [42]. REE concentration ranges (µg/g) in some marine sediments from different oceans across the globe are presented in Table 1.

**Table 1.** Overview of ΣREE range (µg/g) in marine sediments from different oceans, modified after [44].

| Ocean | Matrix | ΣREE Range (µg/g) | Reference |
|---|---|---|---|
| East Siberian Arctic Shelf | Bottom sediments | 104 to 220 | [45] |
| Central North Pacific Ocean | Siliceous sediments | 810.4 | [46] |
| Afanasy Niktin Seamount (ANS) in the Eastern Equatorial Indian Ocean | Cobalt crust | 1727–2511 | [40] |
| Mid-Pacific seamount | Cobalt-rich crusts | 2085 | [47] |
| Indian Ocean | Ferromanganese crust | 928–1570 | [48] |
| Scotia Sea | Ferromanganese crust | 3400 | [49] |
| Eastern South Pacific | Deep-sea mud | 1000–2230 | [50] |
| North Pacific (east and west of Hawaiian Islands) | Deep-sea mud | 400–1000 | |
| Minamitorishima Island in the Western North Pacific | REE-rich Mud | >1446.2 (REE+Y) | [51] |
| South China Sea | Ferromanganese nodule deposits | 1460 (avg) | [52] |
| Indian Ocean | REY-rich mud | >400 | [53] |
| | Marine sediments | 585–920 | |
| Andaman Sea, Indian Ocean | Ferromanganese crust, summit of southern seamount | 1139 | [54] |
| | Ferromanganese crust within two peaks of the same seamount | 2285 | |
| Lakshadweep Sea, Indian Ocean | Ferromanganese crust | La (200) and Y (150) | [55] |
| West Sewell Ridge, Andaman Sea, Indian Ocean | Ferromanganese crust | 1600 | [56] |
| | Manganese nodules | 1186 | |
| Clarion-Clipperton Fracture Zone, North-eastern Pacific Ocean | Deep-sea sediments | >700 | [57] |
| West Clarion-Clipperton Zone, Pacific Ocean | Marine sediments | 454.7 (REE+Y) | [58] |
| North Pacific Ocean near Minamitorishima Island, Japan | Deep-sea mud | >5000 (REE+Y) | [59] |
| Mid Pacific Ocean | Fe-Mn nodules | 1178–1434 | [60] |
| Pacific Ocean | Deep nodules | 1326 | [61] |
| | Shallow nodules | 1398 | |
| Pacific Ocean | ocean-floor sediments | 5000–22,000 (22,000 in selectively recovered biogenic calcium phosphate grains) | [59,62] |

Recent studies suggested that the continental shelf sediments of the Atlantic Ocean and northern South China could potentially become resources for REE, similar to metalliferous deep-sea sediments [63,64]. Sediment samples from the northern South China Sea continental shelf reported concentrations of REE ranging from 32.97 to 349.07 µg/g, with an average concentration of 192.94 µg/g. These concentrations of REE in the bottom sediments of Siberian seas in the Arctic were found to be mainly controlled by the characteristics of the original rocks [64]. Therefore, deep-sea sediments due to their vast amounts of REE and remarkably high proportion of HREE have attracted interest as a future REE resource. In addition, there is a considerable advantage of the mining and metallurgical treatment of these marine sediments, in which the REE and several other metals are not part of the crystal lattice of the minerals that host them, in contrast to the land-based deposits. As a result, the mining and metallurgical treatment has become economically favorable. In addition, the very low Th and U concentrations in these deep-sea deposits do not pose many environmental risks, in contrast to many well-known land-based REE deposits [65]. However, the regulatory authorities are not able to take decisions on future seabed mining due to the possible environmental effects of seabed mining [66].

### 2.3.2. Phosphorites and Related Deposits

Phosphate ores are divided into two main types on the basis of their origin: Sedimentary and igneous phosphate rocks, about 80% of which are of marine origin, and around 17% could be derived from igneous rocks. In addition, the remaining deposits are the residual sedimentary and guano-type deposits [67]. Both igneous and sedimentary phosphate rock deposits are used for manufacturing fertilizers. These rocks contain up to 1% $\sum$REE, and phosphate rock ores; therefore, they are potential future resources of REE. Igneous phosphate rocks (e.g., Kola Peninsula, Russia, and Brazil), and sedimentary phosphate rocks are formed by the deposition of phosphate-rich materials in marine environment and are known as phosphorites, which are found in places, such as Florida, Morocco, and the Middle East [68]. Marine phosphorites concentrate REE, including Y, during the early diagenetic process. The REE content in seamount phosphorites is significantly greater than in continental margin phosphorites [42]. Morocco has the largest phosphate ore reserves in the world with 50 billion tons, followed by China and Egypt with 3.2 and 2.8 billion tons of phosphate ores, respectively [69]. Based on the REE concentration data presented in Table 2, phosphorite deposits can be considered as promising and could represent a profitable alternative source for REE in the future. Buccione et al. [70] made an attempt to understand the economic potential of Northern African phosphorite deposits as alternative REE resources. The very high concentrations of REE (up to 1759 µg/g) in the Northern African phosphorites revealed that they can be considered as promising alternative REE resources.

**Table 2.** Concentrations of REE in marine phosphorites and related rocks from different regions across the world.

| Ocean | Phosphorites | Average Concentration (µg/g) | Reference |
|---|---|---|---|
| Pacific and northeast Atlantic | Seamount phosphorites | 727 ($\sum$REE + Y) | [42] |
| | Continental margin phosphorites | 161 ($\sum$REE + Y) | |
| Doushantou Formation, South China | Danzhai phosphorite deposit | 21 to 447 ($\sum$REE) | [71] |
| Meishucun excavation sites, South China | Cambrian phosphorites | 99.1–709.7 ($\sum$REE) | [72] |
| Sedimentary Abu Tartur phosphate ore, Egypt | Phosphate ore | 0.05–0.20 wt% ($\sum$REE) | [69] |
| Mississippian phosphorites, USA | Phosphorite ore | 18,000 ($\sum$REE) | [73] |

**Table 2.** *Cont.*

| Ocean | Phosphorites | Average Concentration (µg/g) | Reference |
|---|---|---|---|
| Mountain Pass phosphorites, USA | Phosphorite ore | | |
| Chinese clay-type Phosphorite deposits | Phosphorite ore | 500 to 2000 ($\sum$REE) | [74] |
| Hazm Al-Jalamis Phosphorites, Saudi Arabia | Phosphorites | <121.8 ($\sum$REE + Y) | [75] |
| Pabdeh Formation, Khormuj anticline, SW of Iran | Phosphorites | 48 to 682 $\sum$REE | [76] |
| Northern African phosphorite deposits (Morocco, Algeria, and Tunisia) | | 39.2 to 1759.4 $\sum$REE | [71] |
| South China | Phosphorus-bearing dolomites | 330 $\sum$REY | [77] |
| | Phosphorus dolomites | 676 $\sum$REY | |
| | Phosphorites | 1477 $\sum$REY | |

2.3.3. Stream Sediments, River Sediments, and Lake Sediments

If REE-bearing source rocks are present in the catchment area of a river, lake, or stream, then the corresponding sediments continuously accumulate the transported material and become enriched with respect to REE, which is due to the physical and chemical weathering and erosion of source rocks. An increasing trend in the concentration of REE is observed in the sediments starting from river sediments to the estuary and adjacent bay in the case of the Yellow River (Table 3). In sediments, rock-forming accessory minerals, such as monazite, Fe minerals, and clay minerals usually promote the accumulation of REE. Recently, Klein et al. [78] carried out a study on the occurrence and spatial distribution of REE along with a few other elements, such as Ga, Ge, Nb, In, Te, and Ta in the Rhine River sediments and its tributaries in Europe. The Rhine River represents one of the most important waterways in Europe, supporting a large number of industries with a catchment area of around 60 million inhabitants. The $\sum$REE concentrations show a steady increase along the Rhine River (Table 3), inferring that the sources for these elements could be natural as well as anthropogenic activities as the river sediments pose an important sink for anthropogenically introduced REE. Ramesh et al. [79] presented the distribution pattern of REE and several other trace elements in the sediments of the Himalayan rivers. These data are very low as they were obtained more than two decades ago. However, the current concentrations could be considerably higher due to the increased anthropogenic activities along with natural inputs. The lake sediment in Sri Lanka recorded 1101 µg/g of REE (Table 3) since it is in the proximity of the Eppawala phosphate deposit [80]. REE containing source rocks or anthropogenic inputs, or both can be the source for REE in all these sediments.

**Table 3.** REE concentrations in stream sediments, river sediments, and lake sediments in different parts of the world.

| Country | Type of Sediment | $\sum$REE (µg/g) | Reference |
|---|---|---|---|
| Indigirka River, in the Laptev Sea | River sediments | 124 to 197 | [45] |
| Rhine River sediments, Europe | Upper Rhine | 136.07 | [78] |
| | Middle Rhine | 215.32 | |
| | Lower Rhine | 340.45 | |
| | Tributaries | 291.39 | |

**Table 3.** *Cont.*

| Country | Type of Sediment | ∑REE (µg/g) | Reference |
|---|---|---|---|
| Himalayan river system sediments | Brahmaputra | 95 | [79] |
| | Ganges | 97 | |
| | Megna | 107 | |
| | Padma | 131 | |
| | Jamuna | 152 | |
| | Yamuna | 100 | |
| Rivers of the east coast of India | Cauvery | 171 | [81] |
| | Pennar | 203 | |
| | Krishna | 131 | |
| | Godavari-Vasista | 194 | |
| | Godavari-Gauthami | 290 | |
| | Mahanadi | 270 | |
| | Hooghly | 167 | |
| South America | Amazon sediments | 217 | [82] |
| The Mgoua watershed, Cameroon, Africa | Sediments | 282 to 728 Average 550 | [83] |
| South China | Stream sediments | 212 | [84] |
| | Catchment sediments | 187 | |
| | Soils | 190 | |
| Sri Lanka | Lake sediments * | 1011 | [80] |
| Yellow River, China | River sediment | 149 | [85] |
| | Estuary | 165 | |
| | Laizhou Bay | 173 | |

* Lake sediments in the proximity of the Eppawala phosphate deposit in Sri Lanka. Sediment fraction (<63 µm) of each sample.

### 2.3.4. Coal, Coal Fly Ash, and Related Materials

In recent times, coal combustion by-products, including fly and bottom ashes have emerged as potential sources of REE due to their wide availability in abundant quantities. Coal contains more than 200 minerals with about 73 elements, including REE, making it one of the most complex geological materials in nature. All these elements occur in coal in different modes, which are classified into inorganic, organic, and intimate organic associations. In particular, knowing the different modes of occurrence of the critical elements, such as REE is essential for their efficient recovery. REE have been proven to occur in different phases, such as carbonate, silico-phosphate, crandallite group minerals, and apatite phases in coal [86]. Scandium is primarily associated with acid-soluble minerals, while LREE have an association with both acid-soluble minerals and organic complexes, and HREE are dominantly associated with organic matter. Recently, Dai et al. [86] discussed these aspects in-depth. As mentioned above, coal contains trace amounts of REE in the form of discrete minerals or chemically bound to organic matter. Several investigations have demonstrated that coal ash may represent a promising alternative resource of REE. Some investigations revealed that the total REE concentrations in the organic and inorganic phases are 31 and 1141 µg/g, respectively [87]. Coal fly ash is one of the largest industrial waste streams in the world, which contains significant concentrations of REE and has the potential to be an important alternative resource for REE. To date, this is used as a building material for construction, concrete, road base, ceramics, etc. However, currently, several

studies have been initiated for the development of economically feasible procedures for the extraction of REE from coal and coal fly ash as there are large global reserves of coal and coal fly ash worldwide. Yesenchak et al. [88] conducted a study of West Virginia's coal deposits to understand their economic potential on an ash basis with respect to REE and to gain insight into the elemental modes of occurrence and the possible enrichment mechanisms of these elements. Dai et al. [89] have provided an excellent review of anomalous REE and yttrium concentrations in coal. Table 4 presents the potential of these alternative REE resources.

A large country, such as China uses about 4 billion tons of coal for its power plants to generate electricity, which produces about 500–550 million tons of fly ash [90]. Coal and coal combustion products, such as fly ash, bottom ash, and incinerator ash are found to contain significant amounts of REE, as most coal samples contain minor amounts of rhabdophane, Nd, Ce, La $(PO_4)$, $H_2O$, monazite, (Ce, La, Th, Nd)$PO_4$, and xenotime (Y, Er)$PO_4$ [91]. Some of the coal resources are enriched in REE due to the contributions from both detrital and hydrothermal sources [92]. Moreover, REE are probably concentrated in the combustible organic matter of coal (Lin et al., 2017). Significant concentrations of REE found in the global reserves of coal (average REE concentrations are estimated at 380–470 μg/g) and its by-products have prompted numerous research studies to understand their economic feasibility as alternative resources for REE. In addition, this has led to the development of several efficient extraction procedures to extract REE from coal ashes [93]. Therefore, coal fly ash is considered as a potential alternative source for REE in recent years. Moreover, recently, Lu et al. [94] reported that the REE and Y contents of coals from southwestern China are high, and the coal reserves and their by-products are suitable as potential REE sources. For example, in Poland, which is the second largest coal consumer in the European Union, large reserves of fly ash that are rich in REE can be found. The concentrations of REE in certain coal deposits, such as Pond Creek coalbed, Pike County, Kentucky can increase up to 1000 μg/g or more. The collection and chemical analysis of over 4000 samples, including acid mine drainage, coal, and coal ash samples with REE concentrations ranging from ng/mL for raw acid mine drainage materials to thousands of ppm (μg/g) for alternative coal-based resources, such as coal and coal fly ash, USGS [95], recently demonstrated the technical feasibility of producing high-purity critical minerals, including REE from low-grade coal-based materials. Recently, Creason et al. [96] used a geologic and geospatial knowledge-data approach guided by REE accumulation mechanisms to systematically assess and identify areas of higher enrichment with promising potential for different types of coal REE deposits. For a thorough understanding of the origin of anomalous concentrations of REE in coal, in addition to the influence of diagenetic processes, such as temperature, pressure, and time associated with coal-rank advance, it is also necessary to understand a few more interaction mechanisms, such as the influence of marine environments, input of hydrothermal fluids, volcanic ashes, and natural waters on peat swamp [89]. In addition, the presence of considerable amounts of radioactive nuclides ($^{226}$Ra, $^{232}$Th, and $^{238}$U) in fly ash must be considered while developing efficient procedures for the extraction of REE and observing that these hazardous radionuclides should not escape into the environment, leading to radioactive pollution in the surrounding environment [97,98].

**Table 4.** Concentrations of rare earth elements in coal, coal fly ash, and underclay from different sources around the world.

| Place and Country | Material | $\sum$REE (µg/g) | Reference |
|---|---|---|---|
| World average | Fly ash | 450 | [99] |
| | | 404 | [89] |
| Poland | Fly ash | 101–543 | [100] |
| Faer power plant in Guizhou Province, China | Fly ash | 240.20 to 520.27 | [101] |
| lignite coal-based thermal power plants, India | Fly ash | 2100 | [102] |
| Collie Basin, Western Australia | Fly ash | 0.21% $\sum$REO | [103] |
| Pond Creek coalbed, Pike County, Kentucky, US | Coal | <300 to >1000 | [92] |
| World hard coal | Coal | 69 | [89] |
| World low-rank coal | | 65 | |
| World coal | | 68 | |
| US coal | | 62 | |
| China | | 138 | |
| South Korea | Fly ash | 267 to 556 | [104] |
| World | Fly and bottom ash | 0.9–1.3% | [105] |
| Coal bed, Eastern Kentucky, US | Fire clay | 1965–4198 | [106] |
| Qianxi coal-fired power plant, Guizhou province, China | Fly ash | 630.51 | [90] |
| Thermal Power Station II (TS II) of Neyveli Lignite Corporation (NLC), Chennai, Tamil Nadu, India | Fly ash | 2160 ($\sum$REE) 300 (Y) | [107] |
| Central Appalachian Coal-Related Strata, West Virginia (WV) and Central Pennsylvania (PA), US | WV MKT underclay | 297 | [108] |
| | WV MKT coarse coal refuse | 345 | |
| | Central PA LKT underclay | 221 | |
| | Central PA MKT underclay | 728 | |

### 2.3.5. Industrial Waste Products, including Red Mud and Phosphogypsum

Industrial by-products, such as low-grade bauxite, red mud, phosphogypsum, gem mining waste, and slags are found to contain substantial amounts of REE, and thus are considered as alternative resources for REE. In addition to the low-grade bauxites, which are unsuitable for alumina production, high-grade bauxites, especially those in the lowermost parts of karstic-type bauxite deposits, are interesting as an important alternative resource for REE. Numerous studies demonstrated that both low- and high-grade bauxite deposits, such as those on Mediterranean bauxites can be potential resources for REE [12,109–111].

**Table 5.** REE, Y, and Sc concentrations (μg/g) of red mud from different countries in comparison with the Earth's average crust composition, modified from [112].

| Location | La | Ce | Pr | Nd | Sm | Eu | Gd | Tb | Dy | Ho | Er | Yb | Y | Sc |
|---|---|---|---|---|---|---|---|---|---|---|---|---|---|---|
| Average in Earth's crust [113] | 39 | 66 | 9 | 41 | 7 | - | 6 | 1 | 5 | 1 | 3 | 3 | 33 | 22 |
| Chinalco, China [114] | 416 | 842 | 95 | 341 | 64 | - | 56 | 184 | 48 | 25 | 28 | 28 | 266 | 158 |
| Australia [115] | - | - | - | - | - | - | - | - | - | - | - | - | 68 | 54 |
| Brazil [116] | - | - | - | - | - | - | - | - | - | - | - | - | 24 | 43 |
| India [117] | 110 | 70 | 0.5 | - | - | - | - | - | - | - | - | - | 1 | 5 |
| India [118] | 58 | 98 | - | - | - | - | - | - | - | - | - | - | - | 48 |
| India [114] | 112 | 191 | 18 | 48 | 9 | - | 7 | - | 4 | - | 1 | 2 | 13 | 58 |
| Jamaica [119] | 287 | 366 | 74 | 69 | 0 | - | 37 | 0 | 37 | 5 | 21 | 16 | 373 | 55 |
| Greece [120] | 114 | 386 | 28 | 98 | 21 | - | 22 | - | 16 | 4 | 13 | 4 | 75 | 121 |
| Alumine de Greece, Greece [121] | 130 | 480 | 29 | 107 | 19 | - | 22 | 3 | 20 | 4 | 13 | 13 | 108 | - |
| Europe [122] | 151 | 422 | 26 | 121 | 29 | 5 | 23 | - | 14 | 4 | 17 | 16 | 93 | - |
| Greece [109] | 127 | 409 | 28 | 103 | 20 | - | 18 | 2 | 19 | 3 | 11 | 13 | 98 | - |
| Greece [123] | 149 | 418 | 26 | 115 | 29 | 5.0 | 23 | - | 1 | 4.3. | 17 | 16 | 91. | |
| Ajka, Hungary [124] | 114 | 368 | - | - | - | - | - | - | - | - | - | - | 68 | - |
| Turkey [109] | 169 | 480 | 47 | 161 | 32 | - | 4 | 26 | 23 | 4 | 13 | 14 | 113 | - |
| Russian Federation [125] | - | - | - | - | - | - | - | - | - | - | - | - | 53 | 25 |
| Russian Federation [114] | - | - | - | - | - | - | - | - | - | - | - | - | - | 90 |
| Iran [125] | - | - | - | - | - | - | - | - | - | - | - | - | 1 | 19 |
| Slovenia [110] | 182 | 363 | 33 | 116 | 21 | 5 | 19 | 3 | 22 | 5 | 15 | 16 | 131 | - |
| Montenegro (Reebaux (2020) | 292 | 539 | 56 | 208 | 39 | 8 | 35 | 5 | 32 | 7 | 19 | 19 | 174 | - |
| Hungary (Reebaux (2020) | 210 | 429 | 46 | 171 | 32 | 6 | 27 | 4 | 24 | 5 | 14 | 14 | 136 | - |
| Hungary (Coarse material) [110] | 130 | 202 | 28 | 108 | 21 | 5 | 20 | 3 | 18 | 3 | 10 | 10 | 98 | |
| Hungary (Fine material) [110] | 241 | 426 | 54 | 199 | 38 | 7 | 33 | 5 | 29 | 6 | 16 | 16 | 155 | - |

The increasing demand for aluminum for aircrafts, high-speed trains, and building construction worldwide will induce more bauxite mining, and as a result, more low-grade bauxite and red mud will be available for REE extraction in the future. In fact, mining waste in many cases contain high concentrations of toxic substances and is considered as a threat to the environment. However, mining waste discharges produced at the bauxite, phosphate rock, and other ore mines may also become valuable resources, especially for REE. For example, during the process of refining titanium dioxide from the mineral ilmenite, REE oxides are produced as co-products. As a second example, during the leaching of bauxite by NaOH via the Bayer process for alumina production, a solid waste known as red mud is generated, which contains substantial amounts of REE (Table 5). According to a 2011 estimate, the worldwide bauxite residue disposal areas contain an estimated 2.7 billion tons of residue, increasing by approximately 120 million tons per annum [126]. Initially, people used red mud to make bricks and in cement production; however, its potential to be an alternative resource for REE has been recently realized. Recently, Swain et al. [127] revealed that the global production of bauxite and red mud during 2010–2019 is 2999 and 1518 million tons, respectively. China, Australia, Brazil, India, the US, and Russia produce bauxite waste (red mud) in large quantities. Since the inception of Bayer processes, large amounts of red mud have become stockpiled. In addition, with the increasing awareness of the availability of large concentrations of REE in this industrial waste material, this will soon be an important alternative resource for REE [112]. Recent studies revealed that low-grade saprolite ore, which is primarily exploited for its gold and copper, also contains significant amounts of REE [128]. The vast amounts of gem mining industry tailings of past and present mining activities in Sri Lanka have been proven to contain up to 0.3% REE [129]. During the production of wet phosphoric acid and phosphate fertilizer from phosphate ores, a solid phosphogypsum by-product is produced, in which most of the REE (>80%) in the phosphate ore are transferred [69]. Phosphogypsum can show low levels of radioactivity due to the presence of small amounts of uranium and thorium; however, it is also useful as a valuable REE resource (Table 6).

**Table 6.** Concentrations of REE in mine tailings and mine waste of different mines, industrial materials, and industrial waste.

| Country | Industrial Waste | ∑REE (µg/g) | Reference |
|---|---|---|---|
| Poland | Uranium mine tailings, Sudety region | 64.9-109.8 | [130] |
| Southern Shanxi Province, China | Low-grade bauxite | 1539 | [131] |
| Greece | Bauxite | 192 to 1109 (avg. 463) ∑REE + Y+Sc | [132] |
| Australia | Low-grade saprolite ore | 1.14% (∑REE oxides) | [128] |
| Poland | Metallurgical industry waste | >140 | [133] |
| Malaysia | Water Leach Purification (WLP) residue | 88367 with Gd as the most abundant element | [134] |
| Canada | Red Mud | 0.03 wt% | [135] |
| Jamaica | Red Mud | >1303 (REE+Y+Sc) | [119] |
| Alumine de Greece, Greece | Red Mud | >948 (REE+Y) | [136] |
| Turkey | Red Mud | > 1086 (REE+Y) | [109] |
| Europe | Red Mud | 900 | [122] |
| Sri Lanka | Gem Mine | 0.3% (∑REE oxides) | [129] |
| Bagre-Nechí mining district, Colombia | Mine waste (mostly gold mine residue and monazite waste) | 2.19% (Ce, La, Nd, and Pr) | [137] |
| Russia | Different types of red mud | Sc (>100) | [138] |
| Greece | Bauxite residue (Red mud) | 0.1% (∑REE+Y+Sc) | [120] |
| Agios Nikolaos, Greece | Bauxite residue | ∑REE 260 and Sc 120 | [139] |
| Greece | Hematite matrix Bauxite residue | 200 (Sc) 42 to 53 (Sc) | [136] |
| Montenegro | Bauxite residue | 1594 (∑REE) | [111] |
| Europe | Bauxites | <100 to ~500 | [122] |
| SARECO LLP, Kazakhstan | Mineral formations (TMF) from the processing of phosphate uranium ores | 5% | [140] |
| Russia | Phosphogypsum | 0.43–0.52% | [141] |
| Philippine | Phosphogypsum | 266 | [142] |
| Kazakhstan | Titanium-magnesium production waste | - | [143] |

- Not available.

In many instances, industrial waste dumps contaminated with heavy metals pose a large threat to both humans and the ecosystem. Phosphogypsum ($CaSO_4 \cdot 2H_2O$), which is rich in REE, is a waste generated during the production of the phosphoric acid process. In addition, the composition of impurities depends on the source of the phosphate. Phosphogypsum and uranium tailings are being used to extract REE in Poland [130]. These industrial metal wastes can become sources for the extraction of REE. Baron [133] found significant amounts of valuable REE (Table 6) from these waste materials in Poland. For example, Arctic Loparite ore tailings in the Kola Peninsula, North-western Russia are being effectively used to extract REE using bioleaching methods [144].

### 2.3.6. Waste Rock Sources from Old and Closed Mines

Over 100 billion tons of solid waste are produced annually through mining operations all over the world. The use of mine waste and mine tailings is attractive, as this will reduce

mining costs since the material is already excavated from the mine, which would make it easier to access the minerals.

Recently, Abaka-Wood et al. [145] provided a review that highlights the application of new knowledge for the enhanced REE minerals recovery from selected iron ore tailings, silicate-rich tailings, and other alternative resources of similar mineralogical composition.

### 2.3.7. Acid Mine Drainage

Acid mine drainage (AMD) of certain industrial effluents, such as coal, REE, and uranium mines, is also considered as a potential source of REE, although the concentration of ∑REE will be in the ng/mL range in the raw acid mine drainage samples. Table 7 presents concentrations of REE in AMD from different mines across the globe. This is a potential alternative REE resource and the recovery of REE remains largely unexplored. Acid mine drainage is produced by the oxidation of metal sulphides present in some ores and tailings due to the instability of sulphides, which leads to the generation of sulfuric acid that promotes the leaching of the metals that constitute these materials. Depending on the deposit type, acid waters can contain REE in significant concentrations. For example, the acid effluent of a closed uranium mine in Brazil contains 130 ng/mL REE. This acid effluent is treated with lime to produce a precipitate containing 7% of REE [146], although there are other methods, such as ion exchange, solvent extraction, and co-precipitation for their recovery. The treatment of acid mine drainage is often carried out by neutralization, oxidation, and metal hydroxide precipitation. Concentrations of ∑REE in the precipitates obtained after the treatment of the Northern Appalachian Coal Basin in the US, varied from 29 to 1286 μg/g with an average of 517 μg/g [147]. Since REE are present in low concentrations in mining wastewaters and acid mine drainage, adsorbents, such as cellulosic materials and ion exchange materials could be more useful as part of the extraction/preconcentration and/or purification processes. Pyrgaki et al. [148], in a recent review, described mining waste and mine water as promising sources of REE if their extraction is coupled with the simultaneous removal of toxic pollutants. These authors reported very high concentrations of ∑REE+Y from 77 to 1957.7 μg/g in coal combustion ashes, bauxite residue, and phosphogypsum. The ∑REE concentrations in mine discharges from different coal mines range from 0.25 to 9.8 μg/mL, and ore mining areas range from 1.6 to 24.8 μg/mL around the globe.

**Table 7.** Concentrations of REE in acid mine drainage (AMD) from different mines across the globe.

| Name and Location of the Mine | Type | Concentration of ∑REE | Reference |
|---|---|---|---|
| Minas Gerais, Brazil | AMD | 130 ng/mL | [146] |
| Staszic post-mining, Poland | AMD from open pit lake near uranium mine | 993.3 μg/mL | [130] |
| Northern and Central Appalachian Coal Basins, US | Coal mine<br>Treated precipitate<br>Coal mine, Treated AMD | 282 ng/mL<br>517 μg/g<br>724 μg/g | [147,149] |
| Central Appalachian AMD source, US | Pregnant leach solution | 132.02 μg/mL | [150] |
| Sao Domingo mining complex, Iberian Pyrite Belt, Portugal | AMD | <221.8 ng/mL | [151] |
| Xingren coalfield, China | AMD | 118 to 926 ng/mL | [152] |

### 2.3.8. Recycling of E-Waste, Such as Magnets

As the primary and secondary REE ores are becoming depleted, alternative sources, such as the recycling of e-waste for a substantial recovery of REE gained prominence in recent times [153]. Our dependence on REE for the electronic gadgets required for our daily life and green technology is increasing, but the required supply of these elements is far from secure. Due to the environmental problems and the escalating cost of mining, the focus is now shifting toward recycling. Recycling REE to some extent can meet this demand. Currently, electronic waste products, such as magnets, nickel-metal hydride batteries, hard

discs from laptops, desktop computers, mobile phones, loudspeakers, scrap alloy, spent catalysts, and waste light-emitting diodes (LEDs), are ending up in landfills; rather, these electronic devices can be used to recover REE as a mixture of their oxides, which later can be reduced to the production of new devices, such as magnets. More than 50 million tons of e-waste are generated globally every year [154]. Several economically important metals, such as REE, Cu, Co, Li, Sn, Ta, Ga, Au, and Ag can be recovered from this type of electronic waste without affecting the environment [155]. The REE recovery from waste helps sustainable development through the circular economy. Particularly five elements, namely Nd, Eu, Tb, Dy, and Y are extremely important and their demand is expected to grow by 30% in the coming decade. For example, recycling scrap Nd-Fe-B magnets usually provides substantial amounts of Nd, Pr, Dy, and occasionally Tb. Unfortunately, the recycling activity is not heading in the right direction at present mainly since the e-waste contains a very low quantity of these metals. For example, each smartphone contains only 50 mg Nd and 10 mg Pr (Table 8), and in touch screen devices, some of these metals are distributed throughout the material at the molecular scale. On the other hand, large quantities of REE are required for the manufacturing of certain items, such as navy ships, defence aircraft, and permanent magnets. One of the major obstacles in recycling the REE is the complexity of the use of these elements in a variety of applications, with amounts ranging from a few milligrams to a few kilograms [156]. Another critical factor that is affecting the recycling activity of the REE is the lack of cost-effective methods for their extraction from e-waste, both collectively and individually. Therefore, the efficient recovery of these metals becomes entirely difficult and is not cost-effective. Until recently, only about 2% of REE are recovered by recycling processes, compared with 90% for iron and steel [157,158]. Nevertheless, both the US and Japan have intensified their research studies on recycling as the dependence on REE is comparatively high in these countries. Rizos et al. [159] reviewed the recycling of REE permanent magnets from end-of-life (EoL) products and reported that if systematically implemented, this could be among the main avenues for meeting the EU future REE needs and mitigating supply risks.

**Table 8.** Amounts of rare earth elements in different e-waste items.

| E-Waste Item | REE | Concentration per Unit | Reference |
|---|---|---|---|
| Nickel metal hydride (NiMH) battery | $\sum$REE | 5–25% | [160] |
| Cathode-ray tube (CRT) phosphor (as a coating on the panel) | $\sum$REE | 1–7 g | [161] |
| Fluorescent lamp | $\sum$REE + Y | 301.2 mg/1 g phosphor powder | [162] |
| Cathode-ray tube (CRT) | | 265 mg/1 g phosphor powder | |
| Navy submarines | $\sum$REE | 3636 kg | [157] |
| Navy surface ships | $\sum$REE | 1818 kg | |
| Lockheed-Martin F-35 | $\sum$REE | 416 kg | |
| Toyota Prius | $\sum$REE | 15 kg | |
| Air conditioner | $\sum$REE | 120 g | |
| Mobile phone | $\sum$REE | 0.5 g | |
| Wind turbine that generates 3.5 MW Electricity | $\sum$REE | 600 Kg | [1] |
| SmCo$_5$ magnet | Sm | 21.94% | [163] |
| NdFeB magnet | Nd and Pr | 64.5% and 17.32% | [164] |

### 2.3.9. Extra-Terrestrial REE Resources

The Moon is the Earth's nearest neighbor in space and has long been considered as a desirable location for future space mining operations. Earlier studies have demonstrated that the Moon contains vast amounts of natural resources, such as uranium, titanium,

silicon, water, nickel, precious metals, REE, and [3]He at significantly higher concentrations than those found on Earth. Robotic technologies will be extensively utilized for future mining on the Moon to achieve precise localization and conceptual mapping of the lunar surface [165]. Consequently, both government organizations from countries, such as the US and China, as well as private sectors are engaged in space mining activities [166]. Recently, eight nations signed a US-led Artemis Program for Moon exploration and beyond. The path is now clear for these eight nations—Australia, Canada, Japan, Luxembourg, Italy, the UK, the UAE, and the US—to participate in NASA's Artemis program of crewed lunar exploration, which aims to establish a sustainable human presence on and around the Moon by the end of the decade. In addition, several missions are planned for the exploration of the Moon's surface in search of REE and other valuable minerals by countries, such as the USA, China, India, Japan, and Russia [167].

### 3. Conclusions and Future Prospects

In addition to other critical metals, such as Cu, Co, Ga, and Li, REE are required in large quantities for the necessary transition from fossil fuels to green energy solutions in order that the world reaches net zero by 2050. Therefore, there is a great need to intensify the exploration, mining, and extraction efforts for REE not only for the low-carbon energy transition, but also for national security and consumer electronics applications. Moreover, it is time to consider alternative resources for REE since the primary and secondary REE resources are quickly declining and the identification of new resources is becoming difficult. Furthermore, there is great uncertainty regarding the future supply of REE from countries, such as China. The work presented here indicates the range and variety of waste streams with significant REE concentrations and with broader sustainability. Among the alternative resources described here, deep-sea sediments, coal fly ash, and industrial waste products, such as red mud and phosphogypsum are considered as very promising. Future research will focus on finding cost-effective and greener ways of extracting REE from these alternative resources. While developing efficient technologies for the extraction of REE from these alternative REE resources, there is a great need to be careful about the possible emissions of toxic metals, such as Cr and Hg, and radioactive metals, such as U, Th, and toxic organic compounds into the environment. Some recent studies have already established the technical feasibility of a simple, economic, and scalable process for the recovery of REE from most of these alternative REE resources. Moreover, intensive research is currently ongoing worldwide for the development of more efficient technologies for the recovery of REE from these alternative sources, which can demonstrate a solution to the present global REE crisis. Furthermore, some of the recent studies in this direction suggest that other issues, such as environmental, economic, and social factors will strongly influence the development of these alternative REE resources. Finally, there are many questions regarding the ability of alternative sources to become real-world REE sources; however, plausible answers can be obtained only in the future.

**Funding:** This research received no external funding.

**Data Availability Statement:** Not applicable.

**Acknowledgments:** The author would like to acknowledge the support of the Prakash Kumar, Director, CSIR-National Geophysical Research Institute, Hyderabad, India.

**Conflicts of Interest:** The authors declare no conflict of interest.

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
