# Peer review of "Potential Future Alternative Resources for Rare Earth Elements: Opportunities and Challenges"

_minerals, doi:10.3390/min13030425_

Round 1

Reviewer 1 Report

The manuscript "Potential Future Alternative Resources for Rare Earth Elements: Opportunities and Challenges", presents an excellent overview of potential resources for REE extraction/utilization/recovery from primary, secondary and tertiary resources. It is written clearly, comprehensively and contains very well documented  numerous references. The needs of modern society and expected trends in the domains of production and demand of these extremely important critical metals are highlighted.

The author's experience and knowledge in this field is easily noticeable and recognizable, both in conceptualization and methodology, as well as in the selection of references and writing of the manuscript. I believe that the article will be very interesting to Minerals readers.

Besides the above, I have two suggestions for the author regarding minor corrections and additions to the manuscript:

1) Author states: "Industrial by-products like low-grade bauxite, red mud, phosphogypsum, gem mining waste, and slags are found to contain substantial amounts of REE and are hence these 284 are also considered alternative resources for REE" (lines : 283-285, page 9).

In addition to the mentioned "low-grade bauxites"(unsuitable for alumina production - reviewer's note), high-grade bauxites, especially those in the lowermost parts of karstic type bauxite deposits, are also interesting as an important alternative resource for REE. Numerous studies demonstrate  this regarding low and high-grade bauxite deposits, such as those on Mediterranean bauxites. I am using this opportunity to encourage the author to supplement the article with examples about the content and potential of bauxite deposits as alternative resources for REE, for example: Goodenough et al. (2016),  Deady et al. (2014) and references therein.

Deady, D.; Mochos, E.; Goodenough, K.; Williamson, B.; Wall, F. Rare earth elements in karst-bauxites: A novel untapped European resource? In Proceedings of the ERES2014: 1st European Rare Earth Resources Conference, Milos, Greece, 4–7 September 2014; pp. 1–12.

Goodenough, K.M., Schilling, J. Jonsson, E., Kalvig, P., Charles, N. Tuduri, J., Deady, E.A., Sadeghi, M., Schiellerup, H., Müller, A., Bertrand, G., Arvanitidis, N., Eliopoulos, D.G., Shaw, R.A., Thrane, K. &Keulen, N. Europe's rare earth element resource potential: An overview of REE metallogenetic provinces and their geodynamic setting. Ore Geology Reviews, 72. 2016, 838-856.

                Furthermore, I  think that low-grade and high-grade bauxites should be classified in the group of secondary (or rather exogenous) deposits, both in the text and on Fig.1. (page, 3)

                2) It would be advisable to supplement the text of the manuscript and Tab. 5 and 6 (page 11) with data related to the content and potential of REE in bauxites and red mud in ESEE region, from recommended (and other) references:

REEBAUX (2020): Bauxite and bauxite residue as a potential resource of REE in the ESEE region - Booklet (ed. NenadTomašić). KAVA REEBAUX - Prospects of REE recovery from bauxite and bauxite residue in the ESEE region – EIT RM. ISBN 978-953-6076-89-5. University of Zagreb Faculty of Science. p. 86. http://reebaux.gfz.hr/wp-content/uploads/2021/02/REEBAUX-booklet.pdf

Radusinović, S., Papadopoulos, A. (2021): The Potential for REE and Associated Critical Metals in Karstic Bauxites and Bauxite Residue of Montenegro. Minerals 2021, 11, 975.

Author Response

Dear reviewers,

Please kindly check the point to point response in the attachment.

Thanks

Reviewer 2 Report

To be honest, it's an excellent paper for rare earth research with a high vision and profound knowledge. I hope each paper I read could be like this paper in the future.  I will recommend this paper after its publication. I would like to be known by the author and have a chance to talk with him. I think the article could be published now. 

Author Response

Dear reviewers,

Please kindly check the point to point response in the attachment.

Reviewer 3 Report

The paper concerned is a review with huge amount of references. Regarding autocitations, there are 13 with V. Balaram as the first author and 3 as a co-author. One of them is a review on very similar topics from 2019 (nevertheless, most of the literature cited is very new, so I hope that the review presented is atualized enough). I wonder mainly about Ref. No. 5 which, according to title, probably does not deal with REE (or perhaps with some relicts of REE used as catalysts(?) for fabrication of other catalysts).

The text is poorly organized - especially the numbered sections are not split into paragraphs, which is very unpleasant for reading and does not help to  orientate. Some information is repeated two or more times (which is however a little "hidden" so), or isolate sentences are in a section where they are not very suitable; often an information is repeated in the same sentence (perhaps unintentionally).

In several cases I have doubts whether the numeric values or the units used are correct - this holds especially for extremely high REE concentrations in some waters, equivalent to up to almost 1 g/l (note that various units are used for REE concentrations in similar materials).

I would appreciate an information about REE in combustible of coals; I know about some publications (however, relatively old and perhaps difficult to obtain) with high REE values in Czech coals (mainly in the Sokolov basin), however, the high REE contents in coal ash in many examples cited indicate that they cannot be explained by accessory minerals and clay minerals only, and that the coal probably contains REE which were incorporated into organic matter.

Individual remarks (numbers of rows):

28: Promethium does not occur in nature. Scandium is geochemically only partly similar to REE.

29-32: gold and silver occur native especially because they are not easily oxidized. If the same would be true for REE, there would also form natural metallic alloys (although not individual metals).

79: „bastnasite“ – correct: bastnäsite

80: what about Africa? (carbonatites etc.)

102-103: skarn deposits should be mentioned after the magmatic deposits.

108: The sentence seems somewhat incomplete. Perhaps there should be „form“ instead of „from“?

111: „iron and niobium, and REE“ Please, specify the minerals.

116: „…and supply > 95% of the global HREE demand“ – repeated (see r. 108-109)

125: „illuminate“ – correct: ilmenite?

132: „…300 μg/g cut-off grade for mining“ – in black shales, or generally? I would think that 300 ppm is too low concentration for mining (except for loose sediments like beech sands), unless more minerals are used from the rock. Please specify.

137: „…up to 21.76 μg/g“ – that would be really extremely high concentration in water. Please, check the units, and use preferentially μg/l (or equivalent ng/ml, as used below).

141: these concentrations (0.059‒0.547 ng/ml) are by far not high – I guess that the authors analyzed REE for scientific purposes rather than due to economic potential.

146-149: Oil shales should be better placed immediately after black shales, and not after geothermal waters.

156: mass spectrometry, not spectroscopy

159: TESCAN is no particular analytical method but a manufacturer of electron microscopes!

169: ref. 40 is related to phosphate sediments – this information should be added

171: “Northern South China” should be completed to “northern South China Sea”

175: “These elevated concentrations…” what concentrations? Note that the respective values in Tab. 1 are not high, they are typical for fine-grained sediments.

In the whole chapter 2.3.1., the deep-sea and shelf sediments of various types are somewhat chaotically intermixed, it should be ordered (split into at least two paragraphs will help).

Table 1: the last – the value for “ocean-floor sediments” in “Pacific Ocean” 22,000 μg/g (i.e., 2.2 wt. %) is really very high – if the value is correct, the region should be specified (it is surely not representative mean of floor sediments of the whole ocean).

185-186: the formulations should be improved. For example, simple „of marine origin“ is better than „derived from marine origin“; and sedimentary depsits could be „derived from igneous rocks“ as well.

206: the first „or a“ can be replaced by comma

211: and what about the rock-forming accessory minerals like monazite?

215: the formulation „accommodating a large number of industries“ can be surely improved

217: „that.“ (delete the dot)

221: „Since this data is more than two decades, …“ – something is missing

223-224: „Lake sediments“ – what lake? Note that also concentration of REE in the fine fraction (as compared to coarse ones) may contribute to the high values

Table 3: please complement „China“ to South China“ (according to the reference cited)

241: „A of studies…“ – complete

253-254: Please note that REE are probably also concentrated in the combustible organic matter of coal (and then tend to enter the finest ash), e.g. in the Sokolov basic, Czech Republic (as documented by J. Pešek and V. Bouška; these coals are not more mined, but there are surely more cases in the world)

266: is this acid mine drainage from coal mines?

277: potassium is an essential elements – I would not term 40K „hazardous radionuclide“

Table 4: is the „fire clay“ a horizont embedded in coal seam, or what? The fact that also clays associated with coal are presented (the „underclay“…) should be mentioned in table caption.

284: „are hence these“ – word order?

293-294(-295): please check the numbers and correct them, or, at least, improve formulation: „…120 million tonnes and 2.7 billion tonnes of this material was stockpiled“

300-301: why is the sentence about phosphate deposits placed here? Perhaps it belongs a bit further.

304-305: „… 0.3% REE which can become a potential secondary source for many critical raw materials, including REE“ – somewhat repetitive

312-313: metallurgical waste dumps have little to do with phosphogypsum. Perhaps cadmium contained in some sedimentary phosphates is more relevant (to my knowledge, it is separated from the Cd-rich phosphates)

Table 5: the title should be completed to “REE, Y and Sc”

318: the concentrations in Polish mine waste shown in Table 6 are really not high. Is this a specific material that even at low concentration, REE can be extracted?

331-332: „iron-oxide-silicate-rich tailings“ should be specified (silicates are the main rock-forming minerals - almost all mine tailings are silicate-rich)

340: metal sulphides, not „metallic“

344: „130 μg/ml REE“ – really? Please check the units.

348-349: a better formulation: „In the precipitates… the REE concentrations varied…“

355-357: μg/g in as his by far not „very high“ concentration; the concentrations in water would be very high, if correct

Table 7: 993.3 μg/ml – are the units correct? What is „Pregnant leach solution“? (the unit μg/g would rather fit for a solid)

368-670: the sentence „Recycling… currently end up in landfills“ is incomplete.

Table 8: what is „9Nickel“? Is there really 5-25 % of REE in Ni-MH batteries? The concentrations in fluorescent lamp and Cathode-ray tube (around 30 %) are also very high, are they representative? And why is „fly and bottom ash“ presented here?

Conclusions – general comment: is a total transition from fossil to renewable energy sources ecologically meaningful, taking into account, among others, the consumption of trace elements including REE? Perhaps a question for a future discussion.

Author Response

(The authors gave the same response as above.)
